# Effects of Virtual Reality Exercise Program on Blood Glucose, Body Composition, and Exercise Immersion in Patients with Type 2 Diabetes

**DOI:** 10.3390/ijerph20054178

**Published:** 2023-02-26

**Authors:** Yu-jin Lee, Jun-hwa Hong, Myung-haeng Hur, Eun-young Seo

**Affiliations:** 1College of Nursing, University of Eulji, Uijeongbu 11759, Republic of Korea; 2Division of Endocrinology, Department of Internal Medicine, Daejeon Eulji Medical Center, Eulji University School of Medicine, Daejeon 35233, Republic of Korea; 3Department of Nursing, Gyeongbuk College of Health, Gimcheon 39525, Republic of Korea

**Keywords:** virtual reality, exercise, diabetes, blood glucose, fructosamine, body mass index, immersion

## Abstract

Background: This study is a preliminary study to examine the effect of a virtual reality exercise program (VREP) on type 2 diabetes patients. Method: This is a randomized controlled trial for patients with type 2 diabetes (glycated hemoglobin ≥ 6.5%), diagnosed by a specialist. The virtual reality environment was set up by attaching an IoT sensor to an indoor bicycle and linking it with a smartphone, enabling exercise in an immersive virtual reality through a head-mounted display. The VREP was implemented three times a week, for two weeks. The blood glucose, body composition, and exercise immersion were analyzed at baseline, and two weeks before and after the experimental intervention. Result: After VREP application, the mean blood glucose (F = 12.001 *p* < 0.001) and serum fructosamine (F = 3.274, *p* = 0.016) were significantly lower in the virtual reality therapy (VRT) and indoor bicycle exercise (IBE) groups than in the control group. There was no significant difference in the body mass index between the three groups; however, the muscle mass of participants in the VRT and IBE groups significantly increased compared with that of the control (F = 4.445, *p* = 0.003). Additionally, exercise immersion was significantly increased in the VRT group compared with that in the IBE and control groups. Conclusion: A two week VREP had a positive effect on blood glucose, muscle mass, and exercise immersion in patients with type 2 diabetes, and is highly recommended as an effective intervention for blood glucose control in type 2 diabetes.

## 1. Introduction

Diabetes is one of the major chronic diseases, with 463 million people reportedly affected by it worldwide in 2019, which is expected to gradually increase to 578 million by 2030 and 700 million by 2045 [1]. According to the diabetes fact sheet in Korea, 2020, the prevalence of diabetes in South Korea, in individuals over the age of 30 years, has increased from 11.1% in 2013 to 13.8% in 2018. Patients with diabetes often have concomitant obesity, hypertension, and hyperlipidemia, which increases the socioeconomic burden and lowers the quality of life of the individuals; thus, diabetes management is crucial [2]. The main treatment goals for patients with diabetes are to maintain normal blood glucose levels and prevent acute and chronic complications [3]. Treatment is largely divided into drug therapy and lifestyle modification, among which lifestyle modification can be classified into diet and exercise therapy, allowing patients to self-monitor and prevent complications. However, the practice rate of diet and exercise therapy is still quite low [4]. Exercise therapy increases insulin sensitivity, decreases fasting and postprandial blood glucose levels, reduces cardiovascular risk factors and weight, and ensures the well-being of patients [5,6,7,8].

However, only approximately 36% of patients with diabetes in South Korea engage in regular physical activity [4]. In particular, social distancing and isolation measures, attributed to the recent SARS-CoV-2 virus (COVID-19), have further reduced physical activity [9]. Therefore, a new and safe method for encouraging patients with diabetes to continue performing exercise therapy is warranted. Therefore, this study aimed to devise a new exercise therapy for increasing the exercise practice rate of patients, by rapidly exhibiting the exercise benefits, thereby enhancing exercise immersion and providing a short-term intervention effect.

Cycling is a representative aerobic exercise that can be easily performed. In particular, indoor cycling does not require a lot of space, and can be performed at any time regardless of the weather, time, and season, and can be performed in the current COVID-19 situation. It has the advantage of being able to set an appropriate exercise intensity for each individual, by measuring real-time speed, distance, exercise time, and calorie consumption, and by adjusting the resistance and rotation speed of the wheel [10]. However, indoor cycling can be boring, since it is performed alone in a fixed place, which may hinder engagement in continuous and repeated exercise [11]. Applications using virtual reality (VR) are being developed to compensate for these shortcomings, by increasing the interest and fun of indoor exercise and motivating individuals through a sense of achievement [12].

VR is a technology that ensures realistic real-world experience, by stimulating the individual by creating an environment that is difficult or impossible to obtain in reality, using artificial technology [13]. A head-mounted display (HMD), an immersive VR device, is used in games, movies, education, and training, with images and immersive sound, that provide 360° visual immersion. The sense of reality provided by VR enhances exercise ability by sustaining immersion [14], thereby having a positive effect on participation and learning ability, by improving concentration [15]. Recently, in the health care field, treatment using VR has been reported to be effective in improving motor performance, cognitive function, and fall prevention in patients with Parkinson’s disease and stroke [16,17,18,19,20].

However, among the previous studies that applied VR, a scarcity of studies applying VR to patients with type 2 diabetes exists. Therefore, our study attempted to determine the effects on blood glucose, body composition, and exercise immersion, by applying a 2-week VR exercise program (VREP) in patients with type 2 diabetes.

## 2. Materials and Methods

### 2.1. Study Design

This study was a randomized controlled trial, measuring the effects of a 2-week VREP on the blood glucose, body composition, and exercise immersion in patients with type 2 diabetes (Figure 1).

### 2.2. Participants

Patients were recruited via a recruitment notice at the University of Eulji, University Hospital. The inclusion criteria were, patients between 30 and 65 years of age and diagnosed with type 2 diabetes (glycated hemoglobin ≥ 6.5%), who had not participated in any exercise research program in the last 6 months, could use a smartphone, understood the study, and consented to participation. Exclusion criteria were, those with diabetic peripheral neuropathy, diabetic retinopathy, visual impairment, previous lower extremity joint surgery, stroke, severe arthritis, or dizziness.

The sample size was calculated by substituting α value, power, and effect size using the calculation program G-Power 3.1.9.4 (Heinrich Heine University, Germany) [21]. Upon calculating the sample size, through repeated measures analysis of variance (RM ANOVA), with an effect size set at 0.24, significance level at 0.05, power(1-β) at 0.80, number of groups at three each, and correlation coefficient at 0.5, based on a previous study on diabetes [22], the total sample size derived was 39; the study was conducted on 45 participants, considering a dropout rate of 10%. The function of the Microsoft Excel program randomly assigned 15 participants to each of three groups. During the study, one participant in the VR therapy (VRT) group refused to participate in the experiment, owing to dizziness during exercise after wearing an HMD. Two participants in the indoor bicycle exercise (IBE) group dropped out, owing to COVID-19 self-quarantine, and one participant in the control group dropped out, due to hospitalization for surgery. Thus, 14, 13, and 14 participants in the VRT, IBE, and control groups, respectively, were included in this study (Figure 2). 

### 2.3. Experimental Intervention

The exercise program was developed according to the recommendations of the American College of Sport Medicine [23], based on the type and intensity of exercise, after consulting an internal medicine doctor, a professor of nursing, and a sports therapist. An indoor bicycle with easy access to exercise was selected for this study, and the exercise intensity and duration were changed from low to medium intensity, for 40–60 min, considering patients with type 2 diabetes. Since 3 days of resistance exercise and 3 days of aerobic exercise are recommended per week, and the National Academy of Sports Medicine also recommends at least 2 days of resistance exercise and 3 days of aerobic exercise per week [24], our study prescribed 3 days of exercise per week. When exercising with the indoor bicycle, the exercise intensity was set at a low level (gear two, of gears one to ten), which was adjusted to the intensity that would render participants “slightly out of breath” [25]. The exercise program consisted of warm-up, main, and cool-down exercises. First, warm-up stretching was performed before initiating the exercises, which relieved heart and muscle stimulation and improved the exercise capacity, by improving blood flow. The indoor cycling exercise was performed as the main exercise, followed by cool-down stretching, which accelerated the decomposition of lactic acid accumulated in the blood, after the end of the main exercise, to help recover from fatigue [26].

The exercise program was scheduled at a comfortable time for the participants, to ensure exercise three times a week. After checking the respiratory symptoms and fever of the patients, and disinfecting their hands, the CGM Libre sensor was tagged with a smartphone to check their blood glucose. Thereafter, the participants participated in the exercise program according to the explanation and demonstration provided by the authors. After the exercise program, the CGM Libre sensor was tagged with a smartphone to check participants’ blood sugar. 

The VREP was applied to the VRT group for a total of 50 min, which included 10, 30, and 10 min of warm-up, main, and cool-down exercises, respectively; the main exercise consisted of 30 min of VR IBE. The VREP used an indoor bicycle (DP-652-G6, IWHASMP, China), and VR programs and applications.

An internet of things (IoT) sensor was attached to the pedal of an indoor bicycle, converting the indoor bicycle into a VR device. After downloading the VRFit application from Play Store or Apple Store on their smartphone, and upon logging in and connecting to the IoT sensor, the VR background and music were set on the app screen. When the bicycle pedal was turned, following mounting of the mobile phone on the HMD, the set virtual background and music were displayed. The exercise program was applied to the IBE group for a total of 50 min, including 10, 30, and 10 min of warm-up, main, and cool-down exercises, respectively. The control group did not participate in the exercise program, and was allowed to follow their normal daily routine for 2 weeks, without intervention (Appendix A). 

### 2.4. Measurements 

#### 2.4.1. Mean Blood Glucose (MBG)

In this study, the MBG was obtained by attaching a FreeStyle Libre CGM (Abbott Diabetes Care, Alameda, CA, USA) sensor to the upper arm of the participant, and continuously measuring the glucose level through the interstitial fluid. 

#### 2.4.2. Serum Fructosamine 

For serum fructosamine testing, 3 mL of venous blood was collected, placed in a serum separating tube bottle, and sent to the Green Cross for analysis. The test was conducted by a colorimetric method, using Cobas 8000 (c702, Roche Diagnostics, Mannheim, Germany), which was intended to assess short-term average blood glucose level, using the normal range of 205–285 µmol/L as the standard. 

#### 2.4.3. Body Composition

Body mass index (BMI) refers to the value obtained by dividing the weight (kg) of the participants by the square of their height, measured using a body composition analyzer (InBody Dial h20b, Seoul, Korea). The muscle mass (kg) of the participants was measured using a body composition analyzer (InBody Dial h20b, Seoul, Korea). 

#### 2.4.4. Exercise Immersion

Exercise immersion was measured by a sports flow scale, which was developed by modifying the expansion of the Sport Commitment Model scale, developed by Scanlan et al. [27]. The scale comprises 12 items in two domains, of cognitive immersion and behavioral immersion, which were scored on a 5-point Likert scale, ranging from 1 point for “strongly disagree” to 5 points for “strongly agree.” The total score could be as high as 60 points; a higher score indicates a higher level of exercise immersion. The reliability of the scale is Cronbach’s alpha 0.86–0.94, while the reliability in this study was 0.90. 

### 2.5. Data Analysis

The collected data were analyzed using the IBM SPSS software, version 26.0 (IBM Corp., Armonk, NY, USA). The general characteristics of the participants were analyzed by frequency, percentage, and average; the homogeneities of the general characteristics and dependent variables were analyzed by ANOVA and χ^2^-test. To verify the post-effects, ANOVA and RM ANOVA were used. The post hoc analysis was analyzed by Scheffé and least significant difference (LSD) tests. RM ANOVA was performed to test the difference in the effect according to the time change. When the sphericity test result did not satisfy the sphericity, Wilks’ lambda multivariate test was performed for analysis. Partial eta-squared (η^2^) between the groups and time was analyzed, to explain the degree of influence between the three groups. 

### 2.6. Ethical Considerations

Before conducting this study, the research plan was approved by the Institutional Review Board of the University of Eulji (EU21-002). The research was conducted after registration with the Clinical Research Information Service (CRIS) (KCT0006654). The purpose of the study was fully explained to the participants selected for the experiment, before obtaining written consent for their voluntary participation. The possibility of participation and withdrawal from the experiment, premature abandonment, adverse effects, and treatment for such adverse effects, were described and explained in the informed consent form. It was explained to the participants that the collected data would be ID-coded according to the personal information guidelines, and utilized for approximately a year; thereafter, the data would be wiped out by shredding and permanent deletion from the database, after being stored for years. For the VRT group that participated in the study, a gift, exercise equipment, an M2Me IoT sensor, and an HMD were provided in return for participation in the exercise study. The IBE group was provided with gifts and exercise equipment, and the control group was provided with gifts and exercise equipment after completion of the data collection.

## 3. Results

### 3.1. Homogeneity Test for the Participants’ General Characteristics and Previous Dependent Variables

A total of 41 participants were included in the study. The results of one-way ANOVA of the three groups, to verify the previous homogeneity of the general characteristics and dependent variables, are discussed in Table 1. The mean age was 52.93, 49.15, and 53.14 years in the VRT, IBE, and control groups, respectively, indicating no significant difference among the three groups. No significant difference was observed between the three groups based on the length of illness, sex, education level, smoking, and diabetes treatment before the experiment, confirming the homogeneity.

Based on the results of the one-way ANOVA for previous homogeneity of the dependent variables, the homogeneity of the three groups was confirmed, since no significant difference was observed in the MBG measured by CGM, serum fructosamine, BMI, muscle mass, and exercise immersion.

### 3.2. Effects of VREP on the Blood Glucose, Body Composition, and Exercise Immersion

At pre-test (W_2_), the MBG demonstrated no significant difference. At post-test (W_4_), MBG was 122.86 mg/dL, 123.54 mg/dL, and 132.43 mg/dL in the VRT, IBE, and control groups, respectively, indicating no significant difference. Upon analyzing this result using RM ANOVA, a significant difference was observed in the interaction between the group and measurement time (F = 12.001, *p* < 0.001), as shown in Table 2, and the partial η^2^, which was the effect of the VREP according to the group and time, was 0.387 (Figure 3a).

At baseline (W_0_) and pre-test (W_2_), serum fructosamine levels demonstrated no significant difference. At post-test (W_4_), the serum fructosamine level was 298.79 µmol/L, 307.69 µmol/L, and 311.43 µmol/L in the VRT, IBE, and control groups, respectively, indicating no significant difference. Upon analyzing this result using RM ANOVA, a significant difference was observed in the interaction between the group and measurement time (F = 3.274, *p* = 0.016), as shown in Table 2, and partial η^2^, which was the effect of the VREP according to the group and time, was 0.147 (Figure 3b).

At baseline (W_0_) and pre-test (W_2_), the BMI demonstrated no significant difference. At post-test (W_4_), the BMI was 24.94 kg/m^2^, 25.00 kg/m^2^, and 24.85 kg/m^2^ in the VRT, IBE, and control groups, respectively, indicating no significant difference (Figure 3c). 

At baseline (W_0_) and pre-test (W_2_), the muscle mass demonstrated no significant difference. At post-test (W_4_), the muscle mass was 26.48 kg, 28.31 kg, and 24.87 kg in the VRT, IBE, and control groups, respectively, indicating no significant difference. Upon analyzing this result using RM ANOVA, a significant difference was observed in the interaction between the group and measurement time (F = 4.445, *p* = 0.003), as shown in Table 2, and the partial η², which was the effect of the VREP according to the group and time, was 0.190.

At baseline (W_0_) and pre-test (W_2_), the total exercise immersion score demonstrated no significant difference. At post-test (W_4_), the total exercise immersion score was 35.07, 30.00, and 26.14 points in the VRT, IBE, and control groups, respectively, indicating a significant difference among the three groups. Upon analyzing this result using RM ANOVA, a significant difference was observed in the interaction between the group and measurement time (F = 4.418, *p* = 0.004), as shown in Table 2, and the partial η^2^, which was the effect of the VREP according to the group and time, was 0.183 (Figure 3d).

## 4. Discussion

This study evaluated the after effects of implementing a 2-week VREP, on the blood glucose, body composition, and exercise immersion of the participants. Considering the characteristics of patients with diabetes, and the COVID-19 situation, an indoor bicycle incorporating the benefits of a combination of both aerobic and resistance exercises was selected for the exercise intervention method. Additionally, VR was used to increase interest and immersion in the exercise. 

Following the 2-week long experimental intervention, CGM-measured MBG decreased by 15 mg/dL, 9.08 mg/dL, and 0.93 mg/dL in the VRT, IBE, and control groups, respectively. Serum fructosamine decreased by 14.71 µmol/L, 3.31 µmol/L, and 0.07 µmol/L in the VRT, IBE, and control groups, respectively, indicating a significant decrease in the CGM-MBG and serum fructosamine in the VRT group compared with those of the control group. Thus, the level of blood glucose could be decreased by exercising for only 2 weeks. 

Compared to several previous studies evaluating the effect of various types of exercises, such as treadmill and stationary bicycle [28], walking [29], and compound exercises [30,31], as well as a study applying an 8-week long exercise program in patients with type 2 diabetes [32], VREP for 2 weeks appeared to be effective, since the CGM-MBG and serum fructosamine decreased by 15 mg/dL and 14.71 µmol/L, respectively. 

In general, exercise therapy should be performed continuously, and most of the exercise studies in patients with diabetes have been conducted for more than 6 weeks. However, considering the significant decrease in the CGM-measured MBG and serum fructosamine, compound exercise, including aerobic and resistance exercise, for 2 weeks was effective in controlling the blood glucose levels in patients with type 2 diabetes. This study found that exercise, even for a short period of 2 weeks, had a positive effect on blood glucose control in patients with type 2 diabetes, which may motivate patients to start exercising, thereby serving as an attractive point for emphasizing the importance of exercise. 

In the post hoc group analysis, a significant difference was observed in the CGM-MBG in the VRT and IBE groups compared to that of the control group, and in the serum fructosamine between the VRT and control groups. Since CGM-measured MBG is the mean of continuous blood glucose levels, and serum fructosamine is a value that reflects the blood glucose level for 2–3 weeks, VREP was more effective in reducing blood glucose than IBE. Therefore, it is necessary to compare the measurements before and after the experimental intervention for 3 weeks in a future study, to confirm the results of serum fructosamine. 

Considering the effect of VREP on body composition, no significant difference was observed in BMI among the three groups, whereas muscle mass increased by 0.31 kg in the VRT group, 0.26 kg in the IBE group, and decreased by 0.22 kg in the control group. Since in previous studies, 8 weeks of aerobic exercise [33], 12 weeks of walking exercise [34], and 12 weeks of compound exercise [35] decreased the BMI by 1.53 kg/m^2^, 1.75 kg/m^2^, and 1.55 kg/m^2^, respectively, the 2-week intervention period of our study seems insufficient to induce a change in the BMI. A longer period of exercise is required for weight loss and BMI reduction.

Compared to the results of previous studies, conducting resistance exercise for 12 weeks [36], and compound exercise for 12 weeks [37], that reported an increase in muscle mass of 3.4 kg and 0.85 kg, respectively, the increase in muscle mass by 0.31 kg in the VRT group suggests that even 2 weeks of exercise could increase muscle mass. A previous study, using a Theraband resistance band [38], reported that the thickness of both the shoulder muscles started to increase after 2 weeks of exercise intervention; thus, muscle growth could occur just by exercising for 2 weeks. In general, an increase in muscle mass is crucial for healthy body composition; moreover, these results may motivate patients with type 2 diabetes to start exercising.

In the post hoc group analysis, a significant difference in muscle mass in the VRT and IBE group was observed, compared to that of the control group. It is necessary to reconfirm the experimental intervention in this study, by varying the experimental period, owing to the time-dependent effect. 

In this study, owing to the 2-week experimental intervention, exercise immersion in the VRT group was significantly higher than that in the IBE and control groups. This study was conducted among patients with type 2 diabetes, and the symptoms of the participants were checked periodically before and after the experiment, in consideration of the possibility of cybersickness due to the VREP, using an HMD. The average age of the participants in our study was approximately 51.8 years. During the course of this study, a 54-year-old patient in the VRT group dropped out, owing to cyber-sickness, while the remaining participants in the VRT group completed the 2-week experimental intervention. Exercise immersion was much higher in the VRT group than in the IBE group. Therefore, VREP was effective in increasing exercise immersion. 

To date, no studies applying an immersive VREP in patients with diabetes exist. However, VREP implemented in other diseases and age groups was effective in improving physical function and muscle strength, by allowing participants to immerse themselves in the fun of exercise [16,17,18,19,20,39].

The use of VR in certain situations reportedly improves calorie consumption and exercise speed, compared with exercise in daily life [40]. Based on the results of such studies, VR increased exercise immersion and improved exercise effectiveness. Further, VR exercise may have a more positive effect on the patients’ health than IBE alone, which would thereby aid in blood glucose control and health management.

Two weeks after the exercise intervention, the exercise immersion score increased by 7.57 points in the VRT group and 2.46 points in the IBE group, and decreased by 0.29 points in the control group. Based on the post hoc group analysis, a significant difference in exercise immersion was observed between the VRT and IBE group, since the VR exercise intervention was very effective in exercise immersion, and a new exercise program with VR stimulated the interest of the participants and induced motivation for exercise. These results suggest that VR, which provides experiences that cannot be experienced in reality, increases engagement and immersion through interaction with participants, thereby improving satisfaction and giving a sense of achievement [41]. It can contribute to increasing the exercise practice rate, by allowing users to continue the exercise by being immersed in it [42]. In a previous study, when VR was applied to a cycle exercise game, the VR group moved longer distances, with improved immersion, and demonstrated increased physical exercise ability [43]. A VR program using sensors designed to capture bodily movements, could reportedly induce users to actively participate in the experience, by allowing them to easily immerse themselves in the VR world.

The limitations of this study include the small number of participants, short period of application, inability to completely restrict dietary and physical activities, and the fact that continuous monitoring of blood glucose by the CGM device may have psychological effects on blood sugar control and exercise. In addition, one of the disadvantages of immersive VREP is that it can cause cybersickness, and in this study, this resulted in participants dropping out in the middle of the study. Therefore, this was a preliminary study, conducted over a short period of time. Based on the results of this study, it is necessary to conduct another study, that considers the number of subjects, diet, and intervention period, in order to ensure generalizability in the future.

## 5. Conclusions

A 2-week VREP application in patients with diabetes decreased their MBG, increased their muscle mass, and increased exercise immersion. Since the VREP is effective in increasing exercise immersion of the participants, and making them exercise, and even a short period of exercise is effective in reducing blood sugar, it could be a very effective exercise program for patients with diabetes, which could further motivate the participants.

## Figures and Tables

**Figure 1 ijerph-20-04178-f001:**
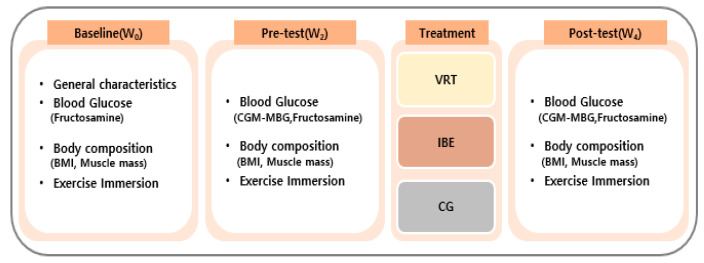
Study design. Baseline (W_0_): baseline measurements. Pre-test (W_2_): measurements two weeks before the intervention. Post-test (W_4_): measurements two weeks after the intervention. VRT: virtual reality therapy group; IBE: indoor bicycle exercise group; CG: control group; CGM-MBG: continuous glucose monitoring mean blood glucose; BMI: body mass index.

**Figure 2 ijerph-20-04178-f002:**
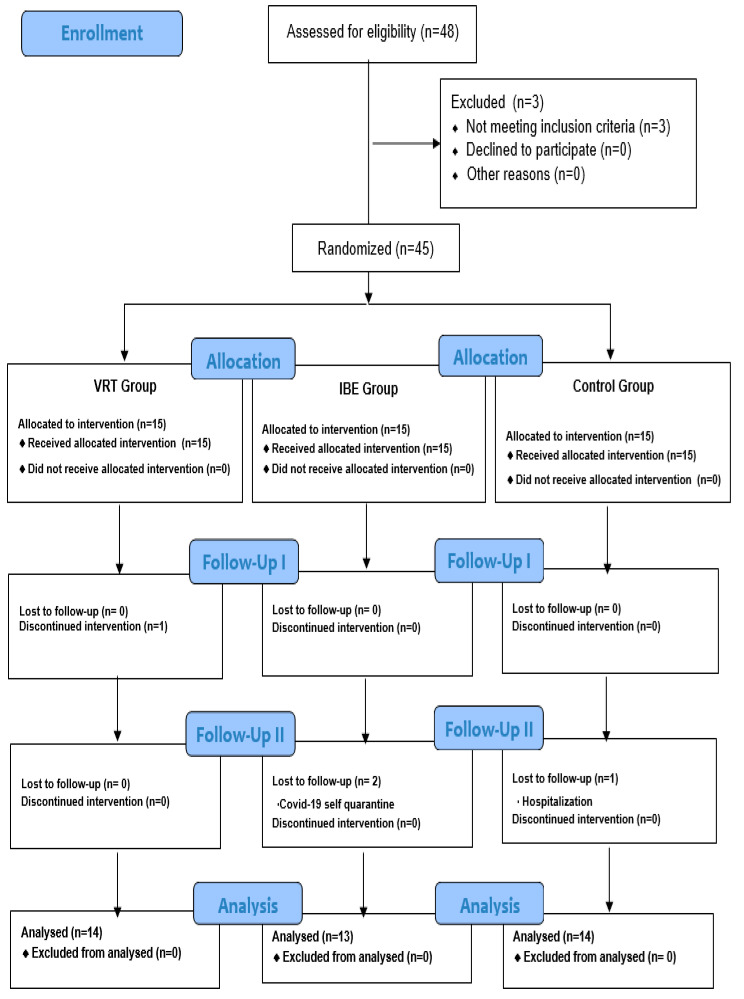
Flow diagram. VRT: Virtual reality therapy group; IBE: indoor bicycle exercise group; CG: control group.

**Figure 3 ijerph-20-04178-f003:**
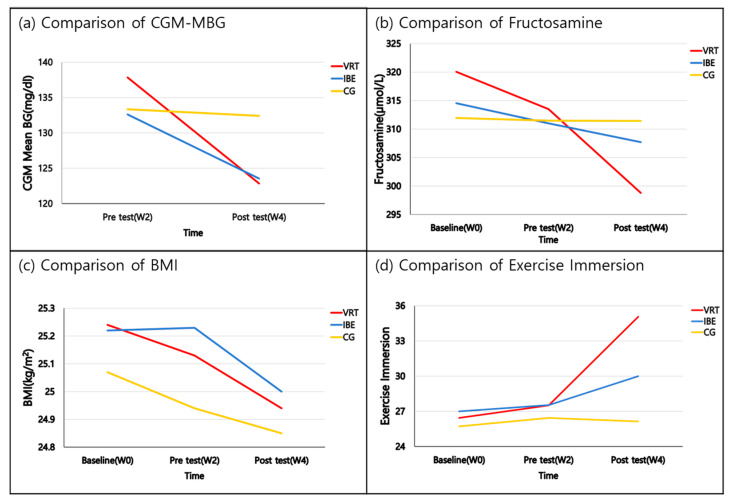
Comparison of blood glucose, body composition, and exercise immersion between the groups. Baseline (W_0_): baseline measurements. Pre-test (W_2_): measurements two weeks before the intervention. Post-test (W_4_): measurements two weeks after the intervention. VRT: Virtual reality therapy group; IBE: indoor bicycle exercise group; CG: control group; CGM-MBG: continuous glucose monitoring mean blood glucose; BMI: body mass index.

**Table 1 ijerph-20-04178-t001:** Homogeneity test for general characteristics and dependent variables between the groups (N = 41).

Variables	Category	VRT (n = 14)	IBE (n = 13)	CG (n = 14)	χ² or F	*p* Value
		Mean ± SD or N (%)	Mean ± SD or N (%)	Mean ± SD or N (%)		
Age (y)		52.93 ± 7.40	49.15 ± 5.15	53.14 ± 7.06	1.518	0.232
Length of illness since onset (m)		57.79 ± 28.25	52.46 ± 34.12	52.71 ± 26.73	0.141	0.869
Gender	Female	7 (50.0)	6 (46.2)	7 (50.0)		
	Male	7 (50.0)	7 (53.8)	7 (50.0)	0.053	0.974
Education	High school	7 (50.0)	4 (30.8)	8 (57.1)		
	More than college	7 (50.0)	9 (69.2)	6 (42.9)	2.000	0.368
Smoking	Yes	1 (7.1)	0	1 (7.1)		
	No	13 (92.9)	13 (100%)	13 (92.9)	0.976	0.614
Admission by DM	Yes	1 (7.1)	0	0		
	No	13 (92.9)	13 (100)	14 (100)	1.977	0.372
Treatment	Oral drug	13 (92.9)	13 (100)	14 (100)		
	Oral drug + insulin injection	1 (7.1)	0	0	1.977	0.372
CGM-MBG (mg/dL)		137.86 ± 31.11	132.62 ± 24.57	133.36 ± 21.84	0.161	0.851
Fructosamine (µmol/L)		320.07 ± 21.72	314.54 ± 41.72	311.93 ± 28.07	0.246	0.783
BMI (kg/m^2^)		25.24 ± 3.14	25.22 ± 2.03	25.07 ± 2.26	0.018	0.982
Muscle mass (kg)		26.18 ± 5.25	28.12 ± 4.70	25.10 ± 4.82	1.284	0.289
Exerciseimmersion		26.43 ± 3.86	27.00 ± 5.49	25.71 ± 9.15	0.130	0.879

VRT: Virtual reality therapy group, IBE: indoor bicycle exercise group, CG: control group. Mean ± SD: mean ± standard deviation, DM: diabetes mellitus, CGM-MBG: continuous glucose monitoring mean blood glucose, BMI: body mass index.

**Table 2 ijerph-20-04178-t002:** Comparison of blood glucose, body composition, and exercise immersion between the groups (N = 41).

Variables		VRT (n = 14)	IBE (n = 13)	CG (n = 14)	F (*p*)	Post Hoc Test	Sources	F (*p*)
CGM-MBG (mg/dL)	Baseline (W_0_)	-	-	-	-		Group Time G*T	0.135 (0.874) 48.878 (<0.001) 12.001 (<0.001)
	Pre-test (W_2_)	137.86 ± 31.11	132.62 ± 24.57	133.36 ± 21.84	0.161 (0.851)			
	Post-test (W_4_)	122.86 ± 24.77	123.54 ± 22.85	132.43 ± 19.50	0.783 (0.464)			
	Difference (W_4_-W_2_)	15.00 ± 10.26 ^a^	−9.08 ± 6.45 ^b^	−0.93 ± 5.15 ^c^	12.001 (<0.001)	a, b > c		
Fructosamine (µmol/L)	Baseline (W_0_)	320.07 ± 21.72	314.54 ± 41.72	311.93 ± 28.07	0.246 (0.783)		Group Time G*T	0.003 (0.997) 7.375 (0.001) 3.274 (0.016)
	Pre-test (W_2_)	313.50 ± 22.10	311.00 ± 28.99	311.50 ± 33.00	0.03 (0.971)			
	Post-test (W_4_)	298.79 ± 27.31	307.69 ± 38.07	311.43 ± 33.15	0.541 (0.587)			
	Difference (W_2_-W_0_)	−6.57 ± 7.40	−3.54 ± 24.32	−0.43 ± 14.73	0.472 (0.627)			
	Difference (W_4_-W_2_)	−14.71 ± 15.35 ^a^	−3.31 ± 13.63	−0.07 ± 16.88 ^c^	3.481 (0.041)	a > c (*p* = 0.053) **		
BMI (kg/m^2^)	Baseline (W_0_)	25.24 ± 3.14	25.22 ± 2.03	25.07 ± 2.26	0.018 (0.982)		Group Time G*T	0.024 (0.977) 6.906 (0.003) 0.835 (0.508)
	Pre-test (W_2_)	25.13 ± 3.02	25.23 ± 2.11	24.94 ± 2.23	0.049 (0.952)			
	Post-test (W_4_)	24.94 ± 2.96	25.00 ± 2.04	24.85 ± 2.09	0.013 (0.987)			
	Difference (W_2_-W_0_)	−0.11 ± 0.24	0.02 ± 0.25	−0.14 ± 0.37	1.037 (0.365)			
	Difference (W_4_-W_2_)	−0.19 ± 0.36	−0.23 ± 0.29	−0.09 ± 0.33	0.725 (0.491)			
Muscle mass (kg)	Baseline (W_0_)	26.18 ± 5.25	28.12 ± 4.70	25.10 ± 4.82	1.284 (0.289)		Group Time G*T	1.408 (0.257) 1.779 (0.176) 4.445 (0.003)
	Pre-test (W_2_)	26.16 ± 5.15	28.05 ± 4.56	25.09 ± 4.80	1.273 (0.292)			
	Post-test (W_4_)	26.48 ± 5.16	28.31 ± 4.58	24.87 ± 4.76	1.695 (0.197)			
	Difference (W_2_-W_0_)	−0.01 ± 0.28	−0.07 ± 0.54	−0.01 ± 0.41	0.087 (0.817)			
	Difference (W_4_-W_2_)	0.31 ± 0.32 ^a^	0.26 ± 0.35 ^b^	−0.22 ± 0.37 ^c^	9.970 (<0.001)	a, b > c		
Exercise immersion	Baseline (W_0_)	26.43 ± 3.86	27.00 ± 5.50	25.71 ± 9.15	0.130 (0.879)		Group Time G*T	3.574 (0.038) 9.741 (<0.001) 4.418 (0.004)
	Pre-test (W_2_)	27.50 ± 3.18	27.54 ± 3.33	26.43 ± 3.86	0.455 (0.638)			
	Post-test (W_4_)	35.07 ± 4.53	30.00 ± 6.23	26.14 ± 4.07	11.242 (<0.001)			
	Difference (W_2_-W_0_)	1.07 ± 5.11	0.54 ± 3.78	0.71 ± 9.75	0.022 (0.978)			
	Difference (W_4_-W_2_)	7.57 ± 6.19 ^a^	2.46 ± 5.84 ^b^	−0.29 ± 1.90 ^c^	8.858 (0.001)	a > b, c		

Baseline (W_0_): baseline measurements. Pre-test (W_2_): measurements two weeks before the intervention. Post-test (W_4_): measurements two weeks after the intervention. VRT: virtual reality therapy group; IBE: indoor bicycle exercise group; CG: control group; CGM-MBG: continuous glucose monitoring mean blood glucose; BMI: body mass index. ** Post hoc test (least significant difference; LSD).

## Data Availability

Not applicable.

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
