# Peer review of "Effects of Virtual Reality Exercise Program on Blood Glucose, Body Composition, and Exercise Immersion in Patients with Type 2 Diabetes"

_ijerph, 2023, doi:10.3390/ijerph20054178_

Round 1

Reviewer 1 Report

I am not familiar with the Sport Commitment Model scale. I cannot evaluate it because I cannot read the paper you introduced. It has not been verified whether the changes in the scale are clinically significant or not.

The primary outcome is blood glucose level, so it is difficult to say clearly that other items are effective.

It is difficult to understand that blood glucose changes purely with or without VR.

How does the author think about whether there is an increase in physical activity during exercise, or an increase in physical activity other than this exercise, or a decrease in food intake?

In table 2, the line between CGM-MBG and Fructosamine is missing and difficult to read.

In table 1, do you measure HbA1c?

Do you assess dietary intake? If not, please describe in LIMITATION.

Author Response

Hello reviewer.

Thank you for your time and passion for my thesis, for reading it carefully and for your valuable and good comments.

  1. The thesis was revised and supplemented according to the reviewer's valuable opinions. I am not familiar with the Sport Commitment Model scale. I cannot evaluate it because I cannot read the paper you introduced. It has not been verified whether the changes in the scale are clinically significant or not.
  • A: I have now attached the papers and questionnaires I referenced. (my apologies because there is no paper translated into English)
  1. The primary outcome is blood glucose level, so it is difficult to say clearly that other items are effective.
  • A: In this study, we investigated blood sugar as the primary outcome, and BMI, muscle mass, and exercise commitment as secondary outcomes. In particular, since this study presented exercise, and it was judged that it is important to continue exercising basically, the results of the BMI, muscle mass, and exercise commitment were also analyzed and presented.
  1. It is difficult to understand that blood glucose changes purely with or without VR.How does the author think about whether there is an increase in physical activity during exercise, or an increase in physical activity other than this exercise, or a decrease in food intake?
  • A: During the experiment, other physical activities could not be restricted according to the research ethics. However, the subjects were selected as those who had no prior participation exercise experience in research before the study and had not engaged in excessive physical activities. Since physical activity improves blood sugar control, it is thought that the blood glucose level of the VRT and IBE groups decreased. (R. J. Sigal, G. P. Kenny, D. H. Wasserman, C. Castaneda-Sceppa and R. D. White, "Physical Activity/Exercise and Type 2 Diabetes: A consensus statement from the American Diabetes Association", Diabetes care, Vol.29, No.6, pp.1433-1438, 2006). In addition to this exercise, physical activity and dietary control have been added as limitations.
  1. In table 2, the line between CGM-MBG and Fructosamine is missing and difficult to read.
  • A: The table has been modified to show them distinctly.
  1. In table 1, do you measure HbA1c?
  • A: As a criterion for the subject selection, HbA1C≥ 6.5% was selected as at the last 3 months.
  1. Do you assess dietary intake? If not, please describe in LIMITATION.
  • A: Dietary intake was recorded in a diary, but strong dietary restrictions could not be applied because the subjects did not live in groups. We have now included this as a limitation.

Again, thank you for giving us the opportunity to strengthen our manuscript with your valuable comments and queries. We have worked hard to incorporate your feedback and hope that these revisions persuade you to accept our submission.

Reviewer 2 Report

Interesting article about the use of immersive virtual reality (IVR) coupled with an indoor bicycle exercise in order to see its effect on blood glucose level, body composition and exercise practice in patients with type 2 diabetes.

Overall, the manuscript is clear and well-structured, with clear and easy-to-interpret figures, and appropriate statistical analysis. In addition, it uses a fairly up-to-date bibliography (almost 50% is from the last 5 years).

However, I would like to make a few comments to the authors.

Major comments:

- The decision to perform an intervention of only 2 weeks duration should with aerobic exercise and expect a significant positive change in a variable such as body mass index should be substantiated (this is something that the authors themselves point out in the Discussion, lines 310-312). On the other hand, there is already literature on the benefits of using the IVR in relation to motivating patients to exercise and positively influencing exercise adherence in samples of older people and also with other types of pathologies. For these reasons, I’m missing a clearer justification for the proposed intervention based on the variables measured.

- It is true that the significant reduction of mean blood glucose and serum fructosamine is an interesting fact (lines 290-296). But it can also lead to false expectations. In addition, the authors make the post-assessment after 2 weeks of the intervention, but there is no further follow-up and we do not know what happens if the proposed exercise is continued or not continued. I think there is a clear weakness or limitation in the study.

- Why do you associate elderly and cybersickness? (lines 327-329 and 335-337) What is the basis for this statement? Does it depend on age or, for example, on the conditions of the intervention performed: speed and variation of the stimuli or IVR time? What is the clinical relevance of a case of cybersickness in the sample? There is a lot of research that shows that the IVR is safe in this population group.

- In relation to articles that support the IVR and the improvement of physical function, there are more recent articles than those indicated by the authors (line 341).

- I think there are more limitations in the article than the authors pointed out.

- I advise changing the wording of the conclusion of the abstract to be more similar to that of the text. It may imply something other than what the results indicate. It would be better to talk about the VREP having a positive effect on muscle mass, rather than on body composition.

Minor comments:

- In the summary there are abbreviations without the corresponding "translation" (VRE, IBE). It also occurs in section 2.5 (LSD). Even if they are known abbreviations, their meaning should be indicated the first time they appear in the abstract and in the text.

- If the abbreviation VRT is used in the text, it would also be advisable to do so in the abstract.

- Tables and figures must be self-explanatory. Figures 2 and 3 lack an explanation of the abbreviations they contain.

Author Response

Review2

Hello reviewer.

Thank you for your time and passion for my thesis, for reading it carefully and for your valuable and good comments.

Major comments:

  1. The decision to perform an intervention of only 2 weeks duration should with aerobic exercise and expect a significant positive change in a variable such as body mass index should be substantiated (this is something that the authors themselves point out in the Discussion, lines 310-312). On the other hand, there is already literature on the benefits of using the IVR in relation to motivating patients to exercise and positively influencing exercise adherence in samples of older people and also with other types of pathologies. For these reasons, I’m missing a clearer justification for the proposed intervention based on the variables measured.
  • A: In this study, VR exercise was provided to diabetic patients, and to confirm the effect of exercise and VR exercise, this study examined the effect of exercise and whether the exercise can be continued. Therefore, blood sugar was examined as the primary outcome, and BMI, muscle mass, and exercise commitment were measured as secondary outcomes. Despite some limitations, VR exercise over a short period of time showed significant results in the interaction between group and time in blood glucose, muscle mass, and exercise commitment variables.

  1. It is true that the significant reduction of mean blood glucose and serum fructosamine is an interesting fact (lines 290-296). But it can also lead to false expectations. In addition, the authors make the post-assessment after 2 weeks of the intervention, but there is no further follow-up and we do not know what happens if the proposed exercise is continued or not continued. I think there is a clear weakness or limitation in the study.
  • A This study was conducted to confirm the effect after 2 weeks of pre-investigation and 2 weeks of experimental treatment, and the effect was confirmed based on four outcome variables: blood sugar, BMI, muscle mass, and exercise commitment. Previous studies confirming the effect of exercise were often conducted over 8 or 12 weeks period; however, was no study has been conducted over a 2-week pre-investigation period, and the effect confirmed over a 2-week period after the experimental treatment as in this study. In this study, in order to present the variable effects before and after the 2-week experimental treatment, the 2-week before and after treatment data were compared. After the experimental treatment, the need for the exercise and continuation, were explained to all the subjects and in the post-response data, there were cases in which exercise was performed more and interest in blood glucose measurement increased. Since the purpose of this study was to analyze the effect after a week of experimental treatment, blood glucose, BMI, and exercise persistence were not measured for all the subjects after the experiment was concluded.
  1. Why do you associate elderly and cybersickness? (lines 327-329 and 335-337) What is the basis for this statement? Does it depend on age or, for example, on the conditions of the intervention performed: speed and variation of the stimuli or IVR time? What is the clinical relevance of a case of cybersickness in the sample? There is a lot of research that shows that the IVR is safe in this population group.
  • A: The subjects of this study were adults with type 2 diabetes, aged between 30 and 65 years, and not the elderly (thank you for pointing out the translation error).
  • In this study, an immersive head mounted display (HMD) was used, and a mobile phone was inserted during use. In previous papers, cybersickness was constantly mentioned when using HMD (Jang, H.J.; Kim, K.H. Study on the influence of VR characteristics on user satisfaction and intention to use continuously -focusing on VR presence, user characteristics, and VR sickness. The Journal of the Korea Contents Association. 2018,18,420-431; Kang, H.G.; Yoo, I.W.; Lee, J.H. Effect of application type on fatigue and visual function in viewing virtual reality (VR) device of Google cardboard type. Journal of Korean Ophthalmic Optics Society 2017,22, 221-228)., Among the subjects in this study, one complained of dizziness and eye fatigue.
  1. In relation to articles that support the IVR and the improvement of physical function, there are more recent articles than those indicated by the authors (line 341).
  • A: References (16-20) have been revised using recent papers.
  1. I think there are more limitations in the article than the authors pointed out.
  • A: We have further modified and supplemented the limitations.

  1. I advise changing the wording of the conclusion of the abstract to be more similar to that of the text. It may imply something other than what the results indicate. It would be better to talk about the VREP having a positive effect on muscle mass, rather than on body composition.
  • A: It has been corrected.

Minor comments:

  1. In the summary there are abbreviations without the corresponding "translation" (VRE, IBE). It also occurs in section 2.5 (LSD). Even if they are known abbreviations, their meaning should be indicated the first time they appear in the abstract and in the text.
  • A: In the abstract and main text, VRE has been changed to VRT, and IBE and LSD was translated and added.
  1. If the abbreviation VRT is used in the text, it would also be advisable to do so in the abstract
  • A: The abstract has been modified by revising VRE to VRT.
  1. Tables and figures must be self-explanatory. Figures 2 and 3 lack an explanation of the abbreviations they contain.
  • A: Corrected, and abbreviations have been added and explained for Figures 2 and 3.

Again, thank you for giving us the opportunity to strengthen our manuscript with your valuable comments and queries. We have worked hard to incorporate your feedback and hope that these revisions persuade you to accept our submission.

Reviewer 3 Report

The study deals with a very interesting and current topic. Immersive exercise in virtual reality offers many benefits. The manuscript correctly describes the experimental procedures, sampling and pre-post evaluations. The small sample is considered adequate given the application of G-power. Blood glucose decreases after two weeks of the experiment. Glucose assessment methods are correct and demonstrate a drop in blood glucose. The programming of motor work is well described. The description of "immersive virtual realities, environment, interface between subjects and environment" could be expanded, but it is a minor detail. The body composition assessment tools give scarcely significant values ​​due to the short duration of the experiment (only two weeks), this is not highlighted. The muscle mass is declared increased even if it is unthinkable that two weeks could have produced a safe and stable increase. In conclusion, the manuscript is well written, clear and on topic. However, the duration of the experiment (2 weeks) does not allow us to approve the scientific nature of the research. It is recommended to present it as an EXPLORATIVE PILOT STUDY "in order to test the effectiveness of the virtual immersive method" before starting a more complete experiment of at least 8<12 weeks. Furthermore, the limited duration of the experiment presented should be indicated in the "limitations of the study", which does not allow for confirmation of the effectiveness of the intervention but represents a first exploratory phase.

Author Response

Review3

Hello reviewer.

Thank you for your time and passion for my thesis, for reading it carefully and for your valuable and good comments.

  1. The study deals with a very interesting and current topic. Immersive exercise in virtual reality offers many benefits. The manuscript correctly describes the experimental procedures, sampling and pre-post evaluations. The small sample is considered adequate given the application of G-power. Blood glucose decreases after two weeks of the experiment. Glucose assessment methods are correct and demonstrate a drop in blood glucose. The programming of motor work is well described. The description of "immersive virtual realities, environment, interface between subjects and environment" could be expanded, but it is a minor detail. The body composition assessment tools give scarcely significant values ​​due to the short duration of the experiment (only two weeks), this is not highlighted. The muscle mass is declared increased even if it is unthinkable that two weeks could have produced a safe and stable increase. In conclusion, the manuscript is well written, clear and on topic. However, the duration of the experiment (2 weeks) does not allow us to approve the scientific nature of the research. It is recommended to present it as an EXPLORATIVE PILOT STUDY "in order to test the effectiveness of the virtual immersive method" before starting a more complete experiment of at least 8<12 weeks. Furthermore, the limited duration of the experiment presented should be indicated in the "limitations of the study", which does not allow for confirmation of the effectiveness of the intervention but represents a first exploratory phase.
  • A: Limitations were added, and according to the reviewer’s opinion, it has been specified in the abstract that this was a preliminary study.

Again, thank you for giving us the opportunity to strengthen our manuscript with your valuable comments and queries. We have worked hard to incorporate your feedback and hope that these revisions persuade you to accept our submission.

Reviewer 4 Report

The authors explore the effects of a VREP on type 2 diabetes patients. The findings of the project are very interesting, as the authors were able to find an improvement in exercise immersion and a positive effect on blood glucose and body composition.

While the improvement in exercise immersion is expected when using immersive VR on the exercise in comparison to traditional methods, the fact that VR use also decreases MBG and increases the muscle mass is very interesting. As stated by the authors, even for a short period of time (2 weeek), the findings can motivate other patients with type 2 diabetes.

The authors could provide more details regarding the VR application used and perhaps some illustrations about it. A link to http://360vrfit.com/ would also help readers know which sensor and HMD were used and could reproduce the experiment morem easily.

The paper is well writen, with only a few minor errors found.

More general comments and a few minor errors found are listed as follows.

" indoor cycling " -> remove line break

"after intervention,." -> "after intervention,"

"respectively were" -> "respectively, were"

"breath [25].”" -> "breath” [25]."

"fro Play Store" -> "from Play Store"

"fora total" -> "for a total"

"2.4.2.." -> "2.4.2."

"USA).The" -> "USA). The"

"groups, respectively " -> "groups, respectively, "

"groups, respectively" -> "groups, respectively,"

please increse the font of the charts in Figure 3

"is be required" -> "is required"

"were effective" -> "was effective"

"Y.J.L,, " -> "Y.J.L, "

Author Response

Review4

Hello reviewer.

Thank you for your time and passion for my thesis, for reading it carefully and for your valuable and good comments.

  1. The authors could provide more details regarding the VR application used and perhaps some illustrations about it. A link to http://360vrfit.com/ would also help readers know which sensor and HMD were used and could reproduce the experiment morem easily.
  • A: Supplement is attached.
  1. More general comments and a few minor errors found are listed as follows.

" indoor cycling " -> remove line break

"after intervention,." -> "after intervention,"

"respectively were" -> "respectively, were"

"breath [25].”" -> "breath” [25]."

"fro Play Store" -> "from Play Store"

"fora total" -> "for a total"

"2.4.2.." -> "2.4.2."

"USA).The" -> "USA). The"

"groups, respectively " -> "groups, respectively, "

"groups, respectively" -> "groups, respectively,"

please increse the font of the charts in Figure 3

"is be required" -> "is required"

"were effective" -> "was effective"

"Y.J.L,, " -> "Y.J.L, "

  • A: Verified and corrected

Again, thank you for giving us the opportunity to strengthen our manuscript with your valuable comments and queries. We have worked hard to incorporate your feedback and hope that these revisions persuade you to accept our submission.

Round 2

Reviewer 1 Report

Well fixed, no problems.

Author Response

Thank you again for your valuable time and feedback. Thanks to you, it was an opportunity for my thesis to grow. I wish you happiness always.

Reviewer 2 Report

I believe that the authors have observed the suggestions and recommendations given by the editor and reviewers, which has benefited the quality of the manuscript. They have also clarified certain aspects, such as updating part of the bibliography, clarifying abbreviations, self-explanatory tables and figures, or better analyzing the limitations of the study. The only point that I think is missing is a clearer and more evidence-based justification for a short intervention such as the one developed in the present study.

Author Response

Hello reviewer.

Thank you for your time and passion for my thesis, for reading it carefully and for your valuable and good comments.

Point 1: I believe that the authors have observed the suggestions and recommendations given by the editor and reviewers, which has benefited the quality of the manuscript. They have also clarified certain aspects, such as updating part of the bibliography, clarifying abbreviations, self-explanatory tables and figures, or better analyzing the limitations of the study. The only point that I think is missing is a clearer and more evidence-based justification for a short intervention such as the one developed in the present study.

Response 1: To be honest we had a lot of concerns about the period from the research design stage.

As in many previous studies, it was thought that if the duration of the experiment was prolonged, the change due to exercise would naturally appear. However, as this study is a kind of preliminary study, it was considered important to quickly find out the relationship between exercise and various variables such as muscle mass, blood sugar, and body mass index.

Referring to the previous study that muscle thickness increased after 2 weeks of exercise, we first planned to find out the change after 2 weeks of exercise. References are attached

(Kim, W.J.; Hur, M.H. Effect of resistance exercise program for middle-aged women with myofascial pain syndrome on shoulder pain, angle of shoulder range of motion, and body composition randomized controlled trial, RCT. J Korean Acad Nurs 2020,50,286–297

 https://doi.org/10.4040/jkan.2020.50.2.286 )

Again, thank you for giving us the opportunity to strengthen our manuscript with your valuable comments and queries. We have worked hard to incorporate your feedback and hope that these revisions persuade you to accept our submission.
